# Nerve Growth Factor and Autoimmune Diseases

Sergio Terracina [1,†], Giampiero Ferraguti [1,†], Luigi Tarani [2], Francesca Fanfarillo [1], Paola Tirassa [3], Massimo Ralli [4], Giannicola Iannella [4], Antonella Polimeni [5], Marco Lucarelli [1,6], Antonio Greco [4] and Marco Fiore [3,*]

1 Department of Experimental Medicine, Sapienza University of Rome, 00185 Rome, Italy
2 Department of Maternal Infantile and Urological Sciences, Sapienza University of Rome, 00185 Rome, Italy
3 Institute of Biochemistry and Cell Biology (IBBC-CNR), Department of Sensory Organs, Sapienza University of Rome, 00185 Rome, Italy
4 Department of Sensory Organs, Sapienza University of Rome, 00185 Roma, Italy
5 Department of Odontostomatological and Maxillofacial Sciences, Sapienza University of Rome, 00185 Rome, Italy
6 Pasteur Institute, Cenci Bolognetti Foundation, Sapienza University of Rome, 00185 Rome, Italy
* Correspondence: marco.fiore@cnr.it
† These authors contributed equally to this work.

**Abstract:** NGF plays a crucial immunomodulatory role and increased levels are found in numerous tissues during autoimmune states. NGF directly modulates innate and adaptive immune responses of B and T cells and causes the release of neuropeptides and neurotransmitters controlling the immune system activation in inflamed tissues. Evidence suggests that NGF is involved in the pathogenesis of numerous immune diseases including autoimmune thyroiditis, chronic arthritis, multiple sclerosis, systemic lupus erythematosus, mastocytosis, and chronic granulomatous disease. Furthermore, as NGF levels have been linked to disease severity, it could be considered an optimal early biomarker to identify therapeutic approach efficacy. In conclusion, by gaining insights into how these molecules function and which cells they interact with, future studies can devise targeted therapies to address various neurological, immunological, and other disorders more effectively. This knowledge may pave the way for innovative treatments based on NGF manipulation aimed at improving the quality of life for individuals affected by diseases involving neurotrophins.

**Keywords:** arthritis; autoimmunity; mastocytosis; multiple sclerosis; neurotrophins; autoimmune thyroiditis; NGF; systemic lupus erythematosus

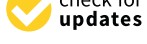



## 1. Introduction

The pathogenic process of autoimmune diseases is intricate and not yet fully understood, and still remains a dearth of research on the involvement of nervous system-mediated immunoregulation [1,2]. Neurotrophins (NTs) represent distinct particles known for their critical function in regulating neuronal development, function and survival [3,4]. Given the diverse functions of NTs in various contexts, any changes or alterations in these molecules can give rise to a range of pathological manifestations associated with different diseases. Nerve growth factor (NGF) stands as the foremost and extensively researched among the NTs, exhibiting activity in a broad array of nervous and non-nervous cell systems. Additionally, NGF is synthesized by various cell types not necessarily linked to NGF-dependent neurons [2,5]. Specifically, NGF plays a vital role in interacting with the immune hematopoietic cell line and assumes an immunomodulatory function primarily by controlling thymic organogenesis and microenvironment [6]. NGF may be also associated with the thymic pathology found in some subtypes of Myasthenia Gravis (MG) as studies have demonstrated that NGF and its receptors are overexpressed in thymic cells of patients affected by MG [7,8]. Moreover, NGF regulates the differentiation and survival of T and B cells [6,9–12]. The blood levels of NGF have been found to be higher in various conditions, including

multiple sclerosis, chronic granulomatous disease, systemic lupus erythematosus, chronic arthritis and mastocytosis [13–15]. Comprehending the role of NTs holds the potential to develop effective and efficient treatment strategies.

The primary objective of this study is to provide a comprehensive summary of the existing literature regarding the involvement of NGF in autoimmune diseases.

## 2. Neurotrophins

The NTs are a family of trophic factors which, despite being initially considered simply survival agents for sympathetic and sensory neurons, have proven to play a major role in controlling crucial traits of development, survival, and the function of neurons in both the central and the peripheral nervous systems [16,17]. NGF is the first NT discovered and analyzed by Rita Levi-Montalcini, Viktor Hamburger and Stanley Cohen more than half a century ago in the early 1950s [18–21]. Other proteins belonging to this family are the brain-derived neurotrophic factor (BDNF) and NTs 3, 4 and 5 [4,22]. NTs are synthesized as pro-NTs with an N-terminal pro-domain and a C-terminal mature domain that subsequently undertake proteolytic cleavage and post-translational changes [23,24]. These premature molecules actually play an active role in being able to complement or alter the function of the complete forms [25]. The final form of NTs binds to high-affinity tropomyosin-related kinase (Trk) A, B or C receptors or the low-affinity p75 pan-neurotrophin receptor (p75NTR), also referred to as CD271 in immune cells [17,26,27]. The TrkA receptor is characterized by the highest affinity for NGF, the TrkB receptor for BDNF and NTs-4/5, the TrkC for NT-3 [28–31]. The NT-3 may link the other Trk receptors with less efficacy. The Trk receptors share a similar structural organization and highly homologous sequence where each Trk receptor extracellular domain consists of a cysteine-rich cluster (C1) followed by three leucine-rich repeats (LRR1–3), two immunoglobulin-like domains (Ig1 and Ig2) and another cysteine-rich cluster (C2) [32–34]. The major site of interaction between NTs and their receptors is the membrane-proximal immunoglobulin-like domain (Ig2). Each Trk receptor crosses the membrane once and ends with a cytoplasmic domain involving of a tyrosine kinase domain enfolded by several tyrosines with the function of phosphorylation-dependent docking sites for cytoplasmic enzymes and adaptors [17].

The p75NTR is a type I transmembrane protein consisting of a transmembrane domain, an extracellular domain, and an intracellular domain. As a member of the tumor necrosis factor receptor superfamily, the p75NTR is able to trigger cell death through the Jun kinase pathways and nuclear factor kappa-light-chain-enhancer of activated B cells (NF-κB) [35]. Actually, p75NTR promotes cell death or survival and modulates neurite outgrowth depending on the operative ligands and co-receptors, so it has been identified as a possible therapeutic target for various diseases [36].

Changes in NTs are associated with a variety of para-physiological states and diseases, including stress situations [37,38], cardiovascular impairments [39,40], cognitive and neurodegenerative disorders [41–44] and pediatric diseases [45,46]. The immunological behavior of diseases associated with NT alteration is often attested by inflammation and organic dysfunction, which could include changes in the *LRRK2* gene [47–49].

Interestingly, Trk receptors and p75NTR can either compete for NGF binding or work together in a cooperative manner, depending on the cellular context [50,51]. The specific combination and balance of these receptors can vary, as the expression of Trk receptors and p75NTR depends on the cell type and the stage of development. For example, during neuronal development, different neurons may express different combinations of Trk receptors and p75NTR based on their specific requirements [3]. In some cases, neurons may co-express both Trk receptors and p75NTR to enable a range of responses to NGF and other NTs. Trk receptors and p75NTR can have both cooperative and competitive interactions in the context of NGF binding [51,52]. When NGF binds to its high-affinity receptor TrkA, it triggers signaling pathways that promote neuronal survival and growth. In some situations, especially during neuronal development or in response to nerve injury, TrkA competes with p75NTR for NGF binding [50,53]. This competitive interaction can influence the balance of

signaling pathways activated by NGF, as p75NTR can mediate signaling pathways distinct from TrkA and may be involved in processes like neuronal apoptosis. On the other hand, in many cells and under certain conditions, both Trk receptors and p75NTR can be present and work concurrently [54,55]. In this cooperative scenario, p75NTR may act to enhance the affinity of Trk receptors for NGF, thus potentiating Trk receptor signaling. This cooperative interaction can adjust the responsiveness of neurons to NGF and other NTs, allowing for more precise control over neuronal development, plasticity, and survival [55].

## 3. Neurotrophins and the Immune System

Figure 1 and Table 1 show the most important evidence for the role of NGF as an immune regulator [2]. NGF expression is increased in inflamed organs, leading to the release of immune-active neuropeptides and neurotransmitters [13].

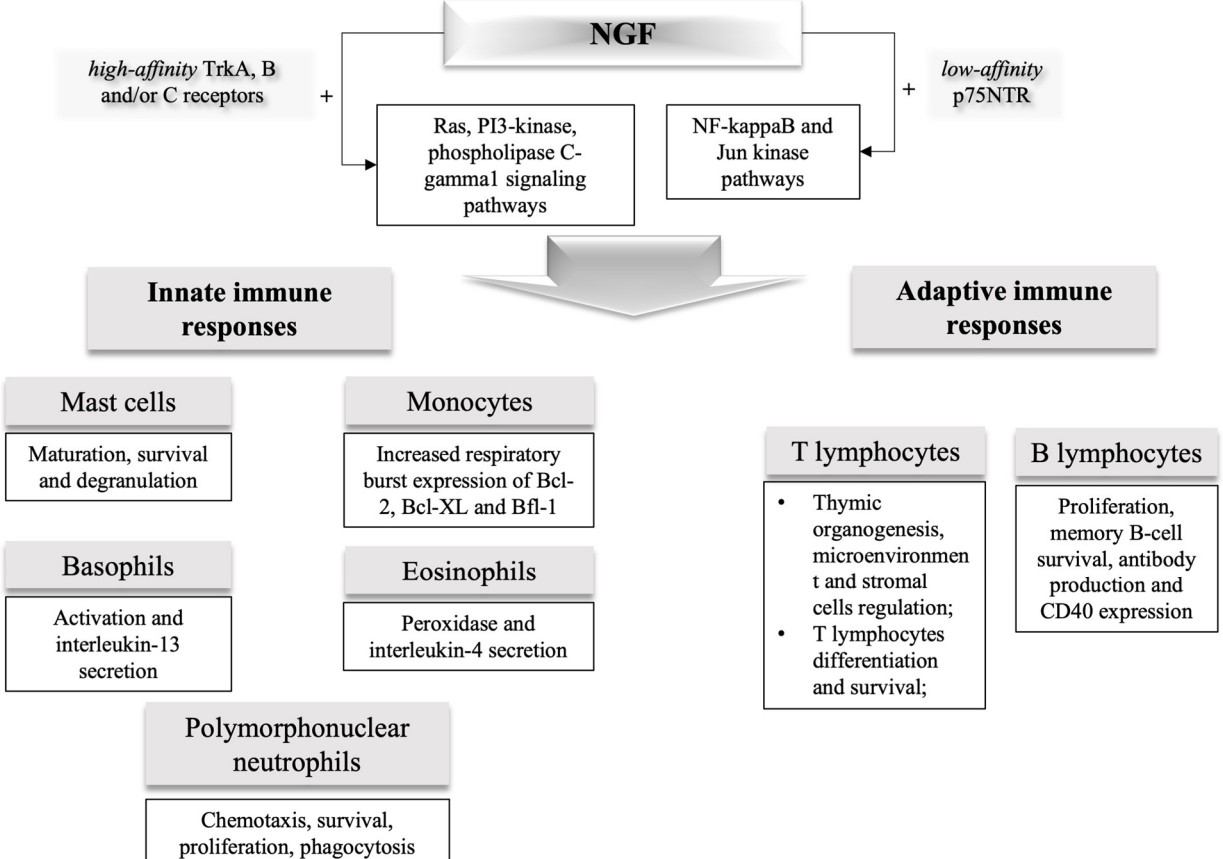

**Figure 1.** Role of NGF as an immune system modulator. Through different pathways (i) high-affinity Trk A, B, and/or C receptors trigger Ras, PI3-kinase, phospholipase C-gamma1 signaling pathways, while (ii) low-affinity p75NTR triggers many paths including those associated with NF-kappaB and Jun kinase. NTs play a major role in many physiological activities including neuroregulation, angiogenesis, immunomodulation, reproduction, and bone tissue regulation. NGF released from tissue mast cells because of nervous, immune, or endocrine stimuli cause innate and adaptive immune responses: mast cells maturation, survival and degranulation; neutrophils chemotaxis, survival, proliferation, phagocytosis; thymic organogenesis, stromal cells, and microenvironment regulation; T-lymphocyte differentiation and survival; B-cell proliferation, memory B-cell survival, antibody production, and CD40 expression; monocytes increased respiratory burst expression of Bcl-2, Bcl-XL, and Bfl-1; eosinophils peroxidase and interleukin-4 secretion; basophil activation and interleukin-13 secretion. NF-kappaB, nuclear factor-kappaB; NGF, nerve growth factor; NTs, neurotrophins; p75NTR, p75 neurotrophin receptor; Trk, tropomyosin-related kinase; PI3, phosphatidyl inositol-3.

**Table 1.** Role of NGF receptor pathways as immune system modulators. Through different pathways activated by its high-affinity (Trk) and low-affinity (p75NTR) receptors, NGF plays a pivotal role in regulating intracellular signaling in various immune cell types. This regulation influences their activation, differentiation, and effector functions depending on the cell type and the context of the immune response. NF-kappaB, nuclear factor-kappaB; NGF, nerve growth factor; p75NTR, p75 neurotrophin receptor; PI3, phosphatidyl inositol-3; Trk, tropomyosin-related kinase; PLC-γ1, phospholipase C-gamma1.

| Receptor | Pathway | Effect on Immune Cells | Ref. |
|---|---|---|---|
| p75NTR | NF-kappaB | 1. B-lymphocytes: B-cell activation and differentiation, regulation of B-cell receptor signaling, antibody production, and B-cell survival.<br>2. T-lymphocytes: activation and proliferation, regulation of cytokine production, immune response, and T-cell survival.<br>3. Mast Cells: release of inflammatory mediators.<br>4. Monocytes: differentiation into macrophages, regulation of production of inflammatory cytokines.<br>5. Basophils: activation and release of proinflammatory molecules.<br>6. Eosinophils: recruitment, activation and production of various cytokines and chemokines.<br>7. Polymorphonuclear Neutrophils: neutrophil chemotaxis, phagocytosis, and production of reactive oxygen species. | [56–66] |
| p75NTR | Jun kinase | 1. B-lymphocytes: B-cell activation, differentiation, and antibody production.<br>2. T-lymphocytes: T-cell receptor signaling and activation, regulation of proliferation, differentiation, and cytokine production.<br>3. Mast Cells: activation and degranulation.<br>4. Monocytes: differentiation into macrophages and their inflammatory responses.<br>5. Basophils: limited role in activation.<br>6. Eosinophils: activation, migration and apoptosis.<br>7. Polymorphonuclear Neutrophils: activation, chemotaxis, phagocytosis and release of reactive oxygen species. | [67–79] |
| Trk | Ras | 1. B-lymphocytes: B-cell receptor signaling and activation, regulation of B-cell proliferation, differentiation, and antibody production.<br>2. T-lymphocytes: T-cell receptor signaling and T-cell activation, proliferation, differentiation and cytokine production.<br>3. Mast Cells: activation and degranulation, release of inflammatory mediators.<br>4. Monocytes: differentiation into macrophages.<br>5. Basophils: activation and the release of inflammatory mediators.<br>6. Eosinophils: activation and survival, chemotaxis, degranulation.<br>7. Polymorphonuclear Neutrophils: chemotaxis, activation, phagocytosis, oxidative burst, and the release of enzymes and antimicrobial proteins. | [80–90] |
| Trk | PI3-kinase | 1. B-lymphocytes: B-cell activation and survival, proliferation and differentiation into antibody-producing plasma cells.<br>2. T-lymphocytes: T-cell activation and survival, proliferation, differentiation into effector subsets, and cytokine production.<br>3. Mast Cells: activation and degranulation.<br>4. Monocytes: activation, migration, and phagocytosis, differentiation into macrophages.<br>5. Basophils: activation, survival, and the release of inflammatory mediators.<br>6. Eosinophils: activation, chemotaxis, degranulation and the release of proinflammatory factors.<br>7. Polymorphonuclear Neutrophils: activation, chemotaxis, and the oxidative burst. | [91–98] |

**Table 1.** *Cont.*

| Receptor | Pathway | Effect on Immune Cells | Ref. |
|---|---|---|---|
| Trk | PLC-γ1 | 1. B-lymphocytes: role in triggering calcium mobilization and activation of downstream signaling pathways, leading to B-cell proliferation and antibody production. <br> 2. T-lymphocytes: crucial for the production of IP3, which leads to calcium release and activation of protein kinase C, promoting T-cell activation and cytokine production. <br> 3. Mast Cells: activation and degranulation. <br> 4. Monocytes: activation, release of inflammatory cytokines and chemokines. <br> 5. Basophils: possible activation and release of inflammatory mediators. <br> 6. Eosinophils: possible activation and release of proinflammatory factors. <br> 7. Polymorphonuclear Neutrophils: PLC-γ1 is not as prominent in neutrophils as in other immune cells. It may play a role in chemotaxis. | [95,99–103] |

Additionally, NGF can directly influence adaptive and innate immune responses. The impact of NTs on immune cells is tightly regulated and may be influenced by other signals [104]. NTs play crucial roles in the microenvironment, stromal cells, thymic organogenesis, and T-cell survival and differentiation [105,106].

NGF is produced, in acquired immunity, not only by the thymus but also by CD4+ T-cell clones [6]. This production induces a cascade of T cell maturation during infection. NTs and their receptors also have significant functions in B cells: NGF may stimulate B-cell proliferation, antibody production, memory B-cell survival, and CD40 expression; BDNF contributes to B-cell maturation in the bone marrow through TrkB95, which is expressed on activated B-cells, and memory B-cells express both TrkA and p75NTR [107]. NTs have also a role in various immunological tumors, particularly B-cell malignancies like acute lymphoblastic leukemia, Burkitt's lymphoma, diffuse large B-cell lymphoma, and multiple myeloma [107–110]. Interestingly, the balance between mature and pro-neurotrophin signaling similarly affects immune function. In fact, increased proNGF/p75NTR signaling in macrophages and glia can alter the functional characteristics of these cells, promoting inflammation and the release of neurotoxic substances [111,112].

During inflammation, mast cells release NGF in high concentrations, leading to axonal outgrowth in nearby nociceptive neurons, resulting in elevated pain perception in inflamed areas. NGF exhibits diverse effects that can be either pro-inflammatory or anti-inflammatory, while the expression of its receptors, TrkA and p75NTR, is dynamically regulated in immune cells, suggesting a variable request for NGF depending on their differentiation and functional activity. This contradictory activity may be explained by the need to limit tissue damage and excessive inflammatory responses: NGF participates in an endogenous mechanism that, while activating immune responses, also triggers pathways necessary to attenuate the inflammatory response and restrain tissue damage [13,47]. Consequently, in patients with inflammatory diseases like those affected by chronic arthritis, reduced immune cell expression of TrkA might lead to a diminished activation of regulatory feedback mechanisms by NGF, thereby contributing to the development and maintenance of persistent inflammation. NGF production is actually stimulated by inflammatory cytokines such as IL-1 and IL-6 in various tissues. Among the numerous cells that produce NGF, T-lymphocytes can be either inhibited or stimulated by this neurotrophic factor, depending on the receptors involved [11,113]. Therefore, it appears plausible that NGF plays a role in finely regulating cellular functions. Furthermore, CD4+ T cells producing NGF might be involved in processes related to neural protection and repair [114]. Although T-cells, including CD4+ T cells, are integral components of the peripheral immune system, it is not common to associate NGF production with peripheral T cell functions. It is worth noting that it remains unclear whether, under certain conditions, peripheral T-cells might also produce NGF if prompted to do so by the local microenvironment or immune signals [6]. However, the production of NGF by T cells appears to be clearly more prominent within

the brain or in specific neurological contexts. Also, B cells autocrinally produce NG which seems to directly impact the synthesis of calcitonin-gene related peptide (CGRP) in B cells, thereby influencing the intensity and duration of the immune response [115].

NTs are naturally produced by our bodies. However, in certain circumstances, antibodies against these molecules may be produced. For example, antibodies against NGF have been discovered in the sera of herpes simplex virus (HSV) patients, suggesting a possible role in modulating NGF's cytokine function during viral infection [116]. In particular, in this study the authors used enzyme-linked immunosorbent assays (ELISA) and immunoadsorbent columns, to detect the presence of NGF-specific autoantibodies that could bind to and immunoprecipitate mouse NGF [116]. Then, the researchers observed higher levels of these anti-NGF antibodies in the blood of patients who were infected with HSV suggesting that the presence of HSV infection is associated with an increase in anti-NGF antibodies in the blood. In contrast to infected patients, rabbits that were intentionally inoculated with HSV did not show an increase in anti-NGF antibodies implying that the immune response to HSV infection in animals may be different from that in humans [116]. Finally, as NGF is known to be involved in promoting HSV latency in vitro, it was suggested that these antibodies could potentially modulate the function of NGF as a cytokine and impact the course of HSV infection [117,118]. It has also been suggested that natural autoantibodies may play a key function as carriers of specific cytokines to target cells. On the other hand, this could also be a case of molecular mimicry where it is possible that the anti-NGF antibodies detected in HSV-infected patients may be the result of the immune system recognizing similar epitopes between HSV and NGF, leading to the production of antibodies that target both the virus and the host protein [117,118].

In fetal animal models, exposure to NGF antibodies leads to significant neuroendocrine impairment, with postnatal manifestations such as atrophied sympathetic and sensory ganglia, smaller thyroid glands, reduced body weight, and sensory deficits [119–121]. Moreover, exposure to NGF antibodies during this developmental period can neutralize NGF in neuroendocrine structures, causing neuroendocrine immunodeficiency syndrome. Autoantibodies to NGF have also been found in the sera of some patients with autoimmune diseases such as systemic lupus erythematosus, autoimmune thyroiditis, and rheumatoid arthritis [122]. Autoantibodies from these pathological cases displayed higher avidity for NGF and a higher poly-reactivity with certain cytoskeletal proteins and DNA compared to those from control human subjects [122].

## 4. Autoimmune Diseases

### 4.1. NGF and Autoimmune Diseases

Table 2 shows the role of NGF in autoimmune diseases. Indeed, evidence indicates that NGF may play a role in the pathogenesis of autoimmune diseases. Studies have revealed elevated levels of NGF in the blood and tissues of individuals affected by autoimmune diseases (thyroiditis, rheumatoid arthritis, and multiple sclerosis). Furthermore, NGF has been shown to modulate immune cell activity and has significant involvement in inflammatory conditions, wherein an increase in NGF, induced by inflammation or stress, might stimulate immune cells and other biological mediators during immunologic insults [15].

A notable aspect is an interaction between NGF and tumor necrosis factor (TNF-$\alpha$), which holds valuable insights into the mechanisms underlying autoimmune inflammatory diseases [14]. Nevertheless, the precise role of NGF in autoimmune diseases remains incompletely understood, necessitating further research to establish the exact relationship between NGF and these conditions. It is important to recognize that NGF is just one of numerous factors that may contribute to the development and progression of autoimmune diseases, as these conditions are intricate and multifactorial. A comprehensive understanding of these factors could potentially lead to novel and effective approaches for managing and treating autoimmune diseases.

**Table 2.** NGF roles in autoimmune diseases.

| Disease | Clinical Manifestations | Role of NGF | Ref. |
|---|---|---|---|
| *Autoimmune thyroiditis* | Hyperthyroidism (e.g., Graves' Disease) and hypothyroidism (e.g., Hashimoto Thyroiditis) with a variety of associated symptoms such as humoral psychotic symptoms, intolerance to cold/hot temperature, weight changes, difficulty in concentration, and eye disorders. | • Role in T cell population homeostasis regulation.<br>• Increased levels of NGF and NGF autoantibodies in the blood and tissues of Autoimmune thyroiditis patients.<br>• NGF may contribute to inflammation and tissue damage stimulating pro-inflammatory cytokines production and activating mast cells.<br>• Anti-inflammatory treatments, able to reduce NGF levels in tears, are able to increase tear film stability and production and decrease eye congestive symptoms. | [122–127] |
| *Chronic arthritis* | Chronic inflammation and damage to joints and surrounding tissues, chronic pain and reduced quality of life, asthenia, psychological and social symptoms | • NGF overexpression in synovial fluid, serum, cerebrospinal fluid, and tissue specimens.<br>• NGF concentrations are correlated with the extent of inflammation and clinical disease activity.<br>• Rapid activation of NF-kB and MAP kinases regulates the bioavailability of aggrecanase and of NGF causing pain.<br>• Decreased TrkA expression in immune cells of arthritis patients may contribute to chronic inflammation development and maintenance by preventing NGF regulatory feed-back mechanisms.<br>• An active proNGF/p75NTR axis promotes chronic synovial inflammation.<br>• Antibodies directed against NGF (NGF-Abs) have been successfully tested for the treatment of chronic pain in both animals and humans with some concerns about side effects. | [13,47,128–137] |
| *Multiple sclerosis* | Periods of relative well-being alternate with episodes of symptom deterioration with gradual worsening over time. Tingling, numbness, pain, burning, itching, reduced sense of touch, loss of strength or dexterity in a limb, vision disorders. | • Increased cerebrospinal fluid and cerebral NGF levels.<br>• Enhanced expression of NGF receptors in multiple sclerosis lesions.<br>• NGF seems to produce anti-inflammatory effects so the induction of NGF probably represents an adaptive response against immune-mediated neuroinflammation.<br>• The release of this neurotrophic factor by brain mast cells could be a key element.<br>• Autocrine and paracrine factor in TrkA-expressing reactive and neoplastic glial cells.<br>• p75NTR plays an important role in leukocyte-endothelial cell interactions and in the maintenance of Purkinje cells survival as well as their upregulation of sodium channel Na(v)1.8.<br>• In animal studies altered NGF levels represent one of the early manifestations of these demyelinating diseases.<br>• Higher levels of NGF correlate with disease phase, duration, age of patients, cognitive performance and disease progression.<br>• Potential therapeutic role as NGF showed neuroprotective activity and immunomodulatory effects.<br>• NGF may be useful as a marker of successful treatment. | [86,89–113] |

**Table 2.** *Cont.*

| Disease | Clinical Manifestations | Role of NGF | Ref. |
|---|---|---|---|
| *Chronic granulomatous disease* | Recurrent infections, multiorgan granulomatous lesions, abscesses, lymphadenitis, hypergammaglobulinemia, anemia. | • The NOX2 deficiency in animal models is linked to a reduced expression of NGF and a decreased generation of mature neurons.<br>• NGF may play a key role in the development of effective therapeutic genetic modification strategies. | [46,138–142] |
| *Systemic lupus erythematosus* | Fever, fatigue, butterfly rash on the face, erythematous lesions in areas exposed to the sun, hair loss, purple-red lesions of the hard and nasal palate, cutaneous vasculitis, and multiorgan involvement. | • Increased B cells and serum NGF levels.<br>• Major role in the inflammatory phase being closely correlated with disease activity.<br>• Higher NGF concentrations related to subcortical atrophy.<br>• NGF may have a prognostic value.<br>• p75NTR is increased on CD16+ and CD56+ leucocytes of affected patients. | [143–151] |
| *Mastocytosis* | Itching, dyspnea, urticaria, dizziness, sense of fainting, multi-organ dysfunctions. | • Mast cells are involved in neuroimmune interactions related to tissue inflammation.<br>• NGF promotes mast cell differentiation and survival while mast cells produce NGF and other neurotrophins.<br>• Elevated serum levels of NGF are related to mast cells load.<br>• Increased expression of modified Trk receptors on mast cells may contribute to the pathophysiology of mastocytosis in paracrine and autocrine loops. | [2,152–154] |

### 4.2. Autoimmune Thyroiditis

Autoimmune thyroid diseases (AITDs) affect approximately 5% of the population and are the most prevalent organ-specific autoimmune conditions [155,156]. These diseases are more common in women, with a prevalence of 5–15%, compared to men with a prevalence of 1–5%. The two most frequent AITDs are Graves' Disease (GD) and Hashimoto Thyroiditis (HT), which are the major causes of hyperthyroidism and hypothyroidism, respectively. Their pathologic features involve reactive T-cells infiltration (predominant in HT) and B cells (predominant in GD), leading to follicular destruction, gradual atrophy, and fibrosis [157].

The etiology of AITDs is multifactorial and involves various factors, including genes like *HLA* and *CTLA4*, as well as smoking, stress, alcohol, and iodine consumption. The onset of injury occurs when autoantibodies and/or sensitized T-cells respond against thyroid cells, causing an inflammatory reaction and cell lysis [158–160]. About 20% of patients with AITDs also have other organ-specific or systemic autoimmune disorders. In the immune system, death receptors from the TNF/NGF receptor superfamily play a crucial role in regulating the adaptive immune response [123]. During the adaptive immune response to antigens, after the peak of the immune response, most activated antigen-specific T cells are eliminated to maintain T cell population homeostasis. This elimination occurs either by death caused by cytokine withdrawal or by activation-induced cell death, through death receptor engagement.

Numerous studies have reported increased levels of NGF and NGF autoantibodies in the blood and tissues of individuals with AITDs [122,124,125]. Additionally, NGF is believed to contribute to inflammation and tissue damage associated with these diseases. It has been found that NGF stimulates the production of pro-inflammatory cytokines and may activate mast cells, which release inflammatory mediators, further contributing to the inflammatory status and tissue damage observed in AITDs. Studies on AITDs-associated ophthalmopathy have emphasized the importance of NGF in the neuroprotection of orbital tissues. This suggests that anti-inflammatory treatments aimed at reducing NGF

levels in tears could enhance tear film stability and production while reducing congestive symptoms [126,127].

Despite these findings, further research is necessary to completely understand the exact association between NGF and autoimmune thyroiditis and its potential implications.

*4.3. Chronic Arthritis*

NGF is overexpressed in numerous inflammatory and degenerative rheumatic diseases [128]. Its presence can be detected in synovial fluid, serum, cerebrospinal fluid, and tissue specimens, with NGF concentrations often correlating with the degree of inflammation and/or clinical activities in various circumstances.

Chronic arthritis is a significant contributor to joint and surrounding tissue inflammation and damage, resulting in chronic pain and reduced quality of life [48]. The upstream mechanism that activates mechanoflammation in chronic arthritis remains unidentified. However, it leads to the rapid activation of NFkB and inflammatory mitogen-activated protein (MAP) kinases, controlling aggrecanase bioavailability and NGF regulation, which in turn causes pain [129]. Numerous studies have demonstrated altered levels of NGF and its receptors in the sera and tissues of patients with chronic arthritis [125,130,131]. The reduced expression of TrkA in the immune cells of arthritis patients may hinder the activation of regulatory feedback mechanisms by NGF, thereby contributing to the development and maintenance of chronic inflammation [13].

Additionally, recent evidence indicates a role for the p75NTR receptor and its preferential ligand proNGF in potentiating inflammatory responses in synovial mononuclear cells of patients affected by chronic arthritis [47]. This suggests that an active proNGF/p75NTR axis may promote pro-inflammatory responses in synovial fibroblasts, further contributing to chronic synovial inflammation. Consequently, p75NTR inhibitors could represent a potential novel therapeutic approach for chronic arthritis. Despite the availability of various non-pharmacologic and pharmacologic treatment options, chronic pain continues to be a significant global burden, affecting approximately 30% of the adult population. Therefore, the development of new analgesics with novel mechanisms of action is of utmost importance. Antibodies targeting NGF (NGF-Abs) have been developed for treating chronic pain conditions such as osteoarthritis and chronic low-back pain, as NGF contributes to peripheral and central sensitization through the activation of TrkA and stimulation of local neuronal sprouting [132,133].

These NGF-Abs have demonstrated significant pain relief and functional improvement in both animal models and clinical patients affected by knee and/or hip osteoarthritis [134,135]. However, their efficacy in non-specific lower back pain has yielded mixed results. Unfortunately, studies have raised safety concerns regarding NGF-Abs, as they may potentially cause or worsen peripheral neuropathies and lead to rapid joint destruction necessitating joint replacement surgery [136,137]. The underlying causes of these side effects have been widely debated, and their pathophysiology remains poorly understood, limiting the practical use of these compounds. Nevertheless, most subjects have shown acceptable tolerability to NGF-Abs, with low rates of discontinuation reported in clinical trials to date [161,162]. Interestingly, research has demonstrated that pretreatment with NGF-Abs reduces or prevents arthritis induced by carrageenan, indicating a functional role of NGF in this type of peripheral inflammation [163].

Similar results have been observed in arthritic transgenic mice expressing high levels of TNF-$\alpha$ in knee joints. Additionally, recent studies on murine models have shown that selective inhibition of TrkA may reduce pain behavior induced by carrageenan or collagen-induced arthritis by inhibiting synovial inflammation. This suggests that NGF blockade is crucial for the beneficial effects (reduction in pain and pathology) in the presence of inflammation [164].

### 4.4. Multiple Sclerosis

Multiple sclerosis is a chronic, predominantly immune-mediated, disease of the central nervous system (CNS) and one of the most common reasons of neurological disability in young adults, characterized by inflammation, demyelination and axonal loss leading to loss of vision in an eye and loss of power or sensibility in an arm or leg [165]. The onset usually begins in young adulthood (between 20 and 40 years of age), and it is more common in women (the female to male ratio is 3 to 1) especially in Europe and North America [166,167]. The etiology is complex and mostly unclear, amongst the environmental factors evidence supports an increased risk in patients with Epstein–Barr virus infection, cigarette smoking, low levels of vitamin D, and an increased BMI during adolescence [168].

In particular, multiple sclerosis is an autoimmune demyelinating disease that produces brain plaques containing mast cells and areas of demyelination demarcated by T-lymphocytes and monocytes in cellular infiltrates [2,169]. NGF seems to produce anti-inflammatory effects by upregulating the production of interleukin 10 by glial cells, T cells infiltrating the CNS, and downregulating the production of interferon-gamma [170]. The major suspected immunogen in multiple sclerosis is the myelin basic protein (MBP) which stimulates mast cell secretion of pro-inflammatory mediators, capable of causing peripheral and central demyelination, and of cytokines that can induce astrocyte production of neurotoxic amounts of nitric oxide (NO) [171]. See Figure 2 for further information.

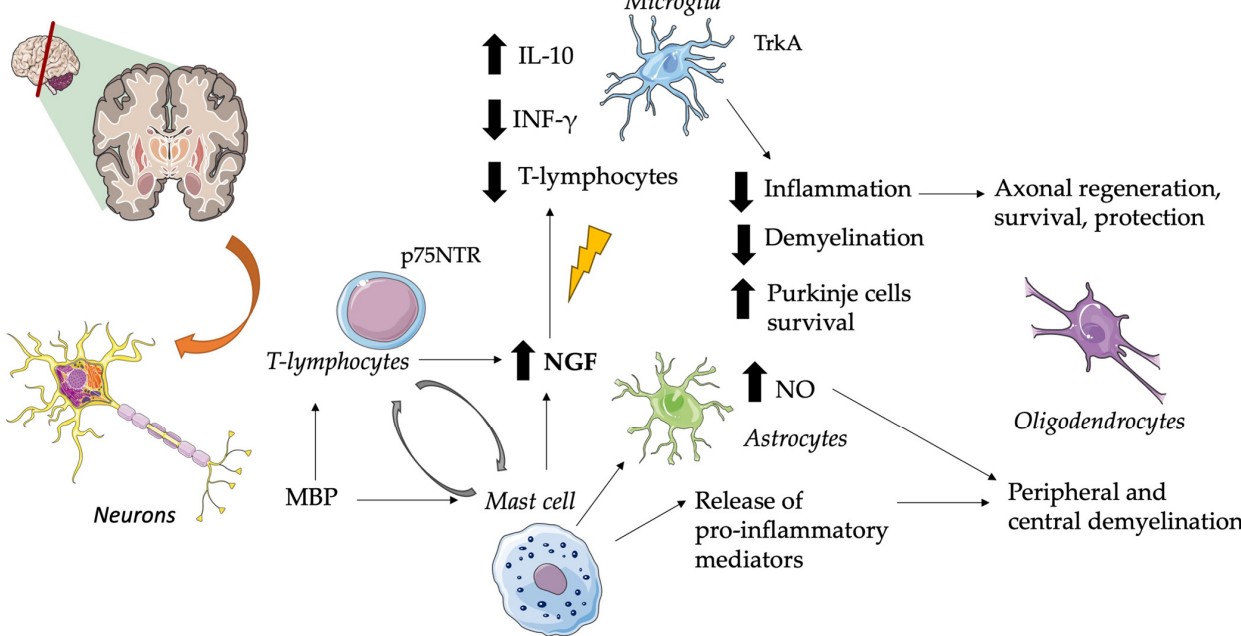

**Figure 2.** Role of NGF in Multiple Sclerosis. Brain plaques characterized by mast cells and demyelination zones containing T-lymphocytes and monocytes are typical of this disease. In this context, NGF exerts anti-inflammatory effects by downregulating the production of IFN-γ, reducing T-lymphocyte infiltration, and upregulating the production of IL-10 by glial cells. This suggests a role as an adaptive response against immune-mediated neuroinflammation. T-cell specific antigens (i.e., MBP) stimulate T-cell and mast cell NGF secretion, but also release pro-inflammatory mediators and astrocyte production of neurotoxic amounts of NO. When p75NTR is expressed, it plays a crucial role in leukocyte-endothelial cell interactions and in maintaining the survival of Purkinje cells. Altered NGF levels are associated with the loss of its neuroprotective role in axonal and oligodendrocyte regulation. CNS, central nervous system; INF-γ, interferon-gamma; IL, interleukin; MBP, myelin basic protein; NO, nitric oxide; NGF, nerve growth factor; p75NTR, p75 neurotrophin receptor; Trk, tyrosine kinase receptor. The pictures were obtained by Servier Medical Art. Servier Medical Art by Servier is licensed under a Creative Commons Attribution 3.0 Unported License (https://creativecommons.org/licenses/by/3.0/ accessed on 5 November 2023).

Early release of mast cell mediators may influence a delayed T-cell response, whereas T-cell products can cause mast cell activation [172]. Since both lymphocytes and monocytes respond to NGF, the release of this neurotrophic factor by brain mast cells could be a key element in such a cycle. It has been suggested that the reactivation of CNS autoimmune T cells by locally presented antigens to which they are specific (e.g., MBP) can lead to enhanced secretion of NTs [173]. So, the induction of NGF probably represents an adaptive response against immune-mediated neuroinflammation [174,175]. There are various reports of increased cerebrospinal fluid and cerebral NGF levels in patients with multiple sclerosis [176–180]. Furthermore, enhanced expression of NGF receptors has also been demonstrated in multiple sclerosis lesions [181,182].

These receptors play important and different roles in multiple sclerosis, for example, NGF acts as an autocrine or paracrine factor in TrkA-expressing reactive and neoplastic glial cells, while p75NTR plays an important role in leukocyte-endothelial cell interactions and in the maintenance of Purkinje cell survival [183,184]. As studies have highlighted, in addition to demyelination and axonal degeneration, dysregulated ion channel expression also contributes to the pathophysiology of multiple sclerosis; moreover, it has been suggested that NGF acts via p75 to contribute to the upregulation of sodium channel Na(v)1.8 in Purkinje cells [185].

In animal studies altered NGF levels represent one of the early manifestations of these demyelinating diseases [186]. Interestingly, a recent article suggested a correlation between higher cerebrospinal fluid levels of iodothyronines, nerve growth factor and multiple sclerosis [187]. This is in line with the evidence suggesting that thyroid hormones activate oligodendrocyte precursors (OPs) and increase myelin-forming protein and NGF content [188]. During the acute phase of the disease, there is an increase of BDNF, TNF-alpha and IFN-gamma synthesis and release, while significantly higher levels of NGF, GDNF, NT3 and NT4 can be found in the post-relapse phase, with the neuroprotective potential of immune cells being inversely related to disease duration and with the age of patients [189]. Interestingly, cognitive performance and disease progression, especially in the case of relapsing-remitting multiple sclerosis patients, are strongly linked to NGF, which might play a neuroprotective role [190,191].

On the other hand, the use of NTs as therapeutic agents has been suggested as a novel option for restoring and maintaining neuronal function during neurodegenerative diseases such as multiple sclerosis. NGF induces axonal regeneration, protection, survival, and differentiation of oligodendrocytes (OGs), it facilitates migration and proliferation of OPs to the sites of myelin damage [192]. NGF also directly regulates key structural proteins that comprise myelin and induces the production of BDNF which is also involved in myelination [192].

As NGF showed neuroprotective activity and immunomodulatory effects, it has been suggested that new therapeutic approaches for the treatment of numerous brain disorders, including multiple sclerosis should focus on NGF and NTs [193,194]. Furthermore, autoimmune and mesenchymal stem cells may protect neuronal populations and suppress the formation of new lesions by the release of NTs, suggesting that these cells could be an alternative source for delivering NTs into the CNS [195].

The NGF and NTs levels are often used as markers of successful treatment in neurodegenerative and autoimmune diseases [196,197]. Interferon beta (INF-β) therapy, which reduces the rate of clinical relapse and the frequency of lesions in patients with multiple sclerosis, has been shown to promote NGF and NT secretion early in the course of this disease, leading to better clinical effects in those patients who presented a significant increase in NTs [198–200].

A recent study demonstrated that six months of probiotic supplementation results in greater improvement in mental health parameters significantly increasing BDNF (but not NGF) levels and reducing the IL-6 levels [201]. Other interesting articles found that moderate exercise training may alter markers of blood-brain barrier (BBB) permeability and neurotrophic factor status, especially in normal-weight persons with multiple sclerosis

influencing the health-related quality of life, while overweight participants may be more resistant to these effects. However, there is still a need for more high-quality studies to clarify the impact of exercise on chronic levels of NTs and long-term health of patients [202–207].

New evidence on the murine model of multiple sclerosis suggests that metformin-induced AMP-activated protein kinase (AMPK) pathway activation stimulates remyelination through induction of neurotrophic factors (NGF, BDNF and ciliary neurotrophic factor), downregulation of neurite outgrowth inhibitor (NogoA) and recruitment of Olig2+ precursor cells opening the way for new therapeutic strategies based on AMPK activation [208].

The anti-inflammatory effects of new drugs and molecules for treating experimental autoimmune encephalomyelitis may provide further insights into the understanding of their neuroprotective activities in multiple sclerosis [209–214]. Unfortunately, most of the evidence on these compounds has not reached the clinical level so their effectiveness on human disease is still unclear.

Finally, it has been highlighted that the importance of NTs and NGF as targets for autoimmune neuroprotection, represents a novel therapeutic approach aimed at shifting the balance between the immune and neuronal cells towards survival pathways in a variety of CNS injuries including multiple sclerosis [215]. These findings are in line with animal evidence that NGF prevents demyelination, cell death, and progression of the disease in experimental allergic encephalomyelitis murine models [216,217]. During the acute phase of the disease, the glial cells become more receptive to NGF, pointing to the glia as an important target for possible pharmacological manipulations such as exogenously administered NGF [218]. In fact, new drugs have been developed that may serve as lead molecules to develop protective agents for oligodendrocyte populations and myelin (NT-like compounds) permeable to the BBB [219].

*4.5. Systemic Lupus Erythematosus*

Systemic lupus erythematosus (SLE) is a rheumatic autoimmune disorder affecting multiple systems, characterized by connective tissue damage due to B-cell hyperactivity and abnormal immune regulation [143]. It is more prevalent in women aged 15 to 40 years, initially manifesting as cutaneous and mucosal erythematous symptoms and photosensitivity. Subsequently, the disease can involve almost all organs and systems, including the kidneys, joints, central nervous system, serous membranes, and hematopoietic system, due to the deposition of immune complexes and complement activation. Patients with SLE have shown increased B cells and higher serum levels of both NGF and BDNF [125,144,145]. Elevated concentrations of NGF and BDNF have been associated with subcortical atrophy in neuropsychiatric SLE patients [146]. NGF also plays a significant role in the inflammatory phase of the disease, and studies have suggested its involvement, along with interleukin-13 (IL-13), in the pathogenesis of SLE, is closely correlated with disease activity [147–149].

Notably, NGF levels have been found to be elevated in childhood SLE, with a correlation to disease activity, indicating its potential role in SLE pathogenesis and its usefulness as a prognostic marker for evaluating disease progression and guiding clinical management [150]. Additionally, higher levels of these factors in SLE patients may be associated with epigenetic changes due to DNA hypomethylation [220]. Interestingly, IL-34, strongly related to myeloid cell subsets (e.g., brain microglia), appears to be associated with disease progression, severity, and chronicity [221,222]. Therefore, blocking NGF, IL-13, and/or IL-34 might be considered to suppress the expression of proinflammatory cytokines in the blood of SLE patients, potentially benefiting the patient's condition.

Despite the valuable evidence in the literature, the role of NGF and its receptors in SLE is still under investigation. Recently, for the first time, the expression of the NGF high-affinity receptor (TrkA) and low-affinity receptor (p75) has been analyzed on all major leukocyte subsets of patients with SLE. When comparing SLE patients with healthy control subjects, TrkA expression was not found to be differentially expressed, while p75 expression was increased on CD16+ and CD56+ leukocytes of patients [151].

### 4.6. Mastocytosis

Mastocytosis is a rare and heterogeneous disease characterized by an increased number of mast cells (MCs) in various body tissues. Two main types of mastocytosis can be distinguished based on their distribution: cutaneous mastocytosis, which is more common in children, and systemic mastocytosis, which primarily affects adults [223]. The clinical features of mastocytosis include flushing, pruritus, abdominal aching, looseness, hypotension, syncope, and musculoskeletal pain [224]. MCs and NGF play a role in neuroimmune interactions associated with tissue inflammation. MCs may produce and respond to NGF, and changes in MCs behavior may lead to altered neuroimmune responses, including autoimmune responses [2,152]. Neurotrophins (NTs) have been found to promote the differentiation and survival of MCs, making them a significant source of NTs [153,154].

Patients with mastocytosis exhibit elevated serum levels of NGF and NT-4, which are related to the load of MCs [2]. Additionally, it has been suggested that the increased expression of modified Trk receptors (TrkA and TrkC) on skin and gut MCs may contribute to the pathophysiology of mastocytosis through autocrine and paracrine loops [153]. Although the precise impact of NGF and its receptors on mastocytosis pathogenesis is not entirely clear, murine models have shown that TrkA activation leads to mastocytosis and is involved in the development of resistance to the receptor tyrosine kinase KIT-targeted therapy, which targets the mast/stem cell growth factor receptor KIT. This suggests that a combined approach targeting both KIT and TrkA might enhance the efficacy of molecular therapy in systemic mastocytosis patients with *KIT* mutations [225]. These findings partially explain why treatment with KIT inhibitors alone has been disappointing in most published clinical trials for mastocytosis.

### 4.7. Chronic Granulomatous Disease

Chronic granulomatous disease (CGD) is a rare disorder causing loss-of-function in the nicotinamide adenine dinucleotide phosphate (NADPH) oxidase (NOX) complex, leading to diminished phagocyte capability in killing microorganisms [138]. Individuals affected by CGD are more vulnerable to infections, excessive inflammation, and autoimmune diseases, as well as experiencing intellectual and cognitive impairment [139,140]. In CGD models with NOX2 deficiency, there is a reduced expression of NTs and a decrease in the generation of mature neurons [138]. NGF plays a significant role in developing effective therapeutic strategies for genetic modification [46].

Most CGD patients are males with hemizygous mutations in the X-linked *CYBB* gene coding for gp91-phox (X-CGD). These patients have significantly low levels of superoxide, as only 5 to 10% of neutrophils producing superoxide are enough to protect X-CGD heterozygotes from severe infections. Recently a promising approach using a bicistronic retroviral vector to modify genetic defects and restore superoxide production in phagocytes of CGD patients has been experimented with offering hope for improving the condition of X-linked CGD individuals. In particular, a potential therapeutic approach for X-CGD involves the development of a retroviral vector containing both the coding sequences of gp91-phox and a cytoplasmically truncated version of human p75NTR [141,142]. Under optimal conditions, this strategy allows 80% of the CD34+ cells to be transduced, resulting in 70% of normal levels of superoxide synthesis and release in phagocytes derived from transduced cells.

## 5. Therapeutic Prospective of Neurotrophins and Their Receptors

Recently, NTs have shown great relevance for their potential role in the therapeutic management of various diseases (immunological disease, neurodegenerative disease, cancer, etc.) [226,227]. The expression of NGF is known to increase in the tissues of patients with immunological diseases and it has been related to severity and treatment efficacy [228,229]. Therefore, NGF normalization has been identified as an optimal target to discriminate therapeutic efficacy in these pathologies [230]. On the other hand, depletion of NGF has

been linked with neurodegenerative disease pathology and symptoms, so replacement strategies have been considered as potential therapeutics [231–233].

Unfortunately, the administration of a therapy involving proteins in the brain has inherent problems because of the blood-brain barrier and many solutions are under study [232,234,235]. Since alterations of NGF and its receptors are common oncogenic drivers stimulating tumor cell survival, migration, proliferation, and invasion, many inhibitors have been produced showing promising therapeutic results [236–238]. These treatments are well tolerated, but some tumors become refractory to this inhibition, so new generations of these therapeutic drugs are being studied [239,240]. Recent evidence suggests that therapies targeting NTs and their receptors may have a major role in various diseases including immunological diseases associated with changes in NGF pathways.

It should be noted that although both autoantibodies against NGF and anti-NGF antibodies involve interactions with NGF, their mechanisms of action and effects differ. The key differences lie in the specificity and purpose of the antibodies. Autoantibodies against NGF are produced by the body's immune system as part of an autoimmune response, and they can disrupt NGF's normal function, potentially leading to neurological and autoimmune conditions [241]. In contrast, anti-NGF antibodies are therapeutic agents designed to block NGF's effects in a controlled manner, without inducing autoimmune reactions, providing relief from pain and inflammation in conditions like arthritis [124,242,243].

The epitopes for these antibodies are indeed different [244]. Autoantibodies against NGF target NGF itself, whereas anti-NGF antibodies are engineered to bind to specific sites on NGF, preventing it from interacting with its receptors. This targeted binding allows anti-NGF antibodies to reduce pain and inflammation while minimizing the risk of autoimmune complications for conditions associated with excessive pain, such as osteoarthritis and rheumatoid arthritis [245].

## 6. Discussion

In this narrative review, we discuss scientific evidence on the role of NGF in autoimmune diseases. Under the influence of various signals, NTs, and in particular NGF, play a crucial immunomodulatory role in mediating the release of immune-active neuropeptides and neurotransmitters, also directly influencing innate and adaptive immune responses [13]. NGF plays a major role in both T- and B-cell differentiation and survival [105,106]. In particular, NGF stimulates B-cell proliferation and antibody production so that its upregulation in inflammatory and immune diseases has been linked to more severe clinical presentation [107]. Furthermore, NGF normalization has been identified as an optimal index to evaluate therapeutic efficacy in immune and inflammatory diseases [230].

The higher levels of NGF in patients with inflammatory diseases have been partially linked to decreased immune cell expression of TrkA which might reduce the activation of regulatory feedback mechanisms by NGF, thus contributing to the development and maintenance of persistent inflammation. Evidence suggests that NGF is involved in the pathogenesis of numerous immune diseases including autoimmune thyroiditis, chronic arthritis, multiple sclerosis, systemic lupus erythematosus, mastocytosis and chronic granulomatous disease. Genetic mutations affecting the production of NGF, or mutations in the receptors Trk and p75NTR, may potentially play a role in autoimmune diseases, particularly through their impact on the immune system and inflammation. Unfortunately, further research is required to elucidate specific mutations and their mechanisms, as well as the finer details of the role of NTs in autoimmune diseases.

## 7. Conclusions

Extensive research conducted over the past decades has revealed the crucial role of NGF in maintaining immune homeostasis, with its activities deeply interconnected across various systems. As we move forward, it becomes imperative for further studies to encompass the intricate and interactive aspects of NGF physiology. By deciphering the specific signaling cascades in which NGF is involved and understanding its precise pathological

contributions, we can pave the way for innovative therapeutic approaches. Manipulating NGF's intracellular pathways holds promise for developing targeted interventions that could revolutionize the treatment of various conditions. In this pursuit, a comprehensive understanding of NGF's multifaceted functions will be instrumental in unlocking its full potential as a therapeutic target.

**Author Contributions:** Conceptualization, S.T., G.F. and M.F.; methodology, S.T., G.F., P.T., M.L., F.F., A.P. and M.F.; validation, L.T., M.R., G.I. and A.G.; formal analysis, S.T., G.F. and M.F.; resources, S.T., G.F. and M.F.; data curation, L.T., M.R., G.I. and A.G.; writing—original draft preparation, S.T.; writing—review and editing, S.T., G.F. and M.F.; supervision S.T., G.F., P.T., M.L., F.F., A.P., M.F., L.T., M.R., G.I. and A.G. All authors have read and agreed to the published version of the manuscript.

**Funding:** This research received no external funding.

**Institutional Review Board Statement:** Not applicable for reviews works.

**Informed Consent Statement:** Not applicable for reviews works.

**Acknowledgments:** Authors thank Sapienza Università di Roma, Italy, and IBBC-CNR of Rome, Italy.

**Conflicts of Interest:** The authors declare no conflict of interest.

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
