# Peer review of "Nerve Growth Factor and Autoimmune Diseases"

_cimb, doi:10.3390/cimb45110562_

Round 1

Reviewer 1 Report

Comments and Suggestions for Authors

This is a very useful, interesting and considerable review aimed at clarifying the dual role of NGF in modulating the immune response involved in the pathogenesis of autoimmune diseases, paving the way for innovative therapeutic approaches and targeted interventions.

I really appreciate the "scientific" search strategy adopted by the authors to carry out the search, cross-referencing the relevant keywords and defining the inclusion criteria.

Just one important issue:

adequate iconographic support is missing.

The authors have provided a table summarizing the main points of involvement of NGF in those autoimmune diseases that they have selected as most relevant; but, at least for Multiple Sclerosis, which is the most analyzed in the review (more than 25% (50/174) of the references are included in the MS paragraph), a Figure should be added.

Minor point:

the authors cited the role of NGF in controlling thymic organogenesis and microenvironmentin ; is there, to the best of their knowledge, any research relating NGF to Myasthenia gravis with anti-AChR antibodies? This subtype of MG is known to be characterized by a strong association with thymic pathology.

Author Response

Answers to the comments raised by Reviewer 1

This is a very useful, interesting and considerable review aimed at clarifying the dual role of NGF in modulating the immune response involved in the pathogenesis of autoimmune diseases, paving the way for innovative therapeutic approaches and targeted interventions.

I really appreciate the "scientific" search strategy adopted by the authors to carry out the search, cross-referencing the relevant keywords and defining the inclusion criteria.

Reply: we thank the reviewer for the positive comments. Indeed, according to the comments of the 2 reviewers we made changes highlighted in light yellow. Furthermore, we included an additional Figure and an additional Table.

Just one important issue:

adequate iconographic support is missing.

The authors have provided a table summarizing the main points of involvement of NGF in those autoimmune diseases that they have selected as most relevant; but, at least for Multiple Sclerosis, which is the most analyzed in the review (more than 50% of the references are included in the MS paragraph), a Figure should be added.

Reply: as requested, we added a Figure on NGF and Multiple Sclerosis section (Figure 2 of the revised paper)

Minor point:

the authors cited the role of NGF in controlling thymic organogenesis and microenvironment; is there, to the best of their knowledge, any research relating NGF to Myasthenia gravis with anti-AChR antibodies? This subtype of MG is known to be characterized by a strong association with thymic pathology.

Reply: as suggested, we added some evidence on NGF and its role in causing thymic pathology in some subtypes of Myasthenia Gravis (MG), unfortunately, there is no specific major information on MG with anti-AChR antibodies (page 2, lines 50-53 of the revised paper).

Reviewer 2 Report

Comments and Suggestions for Authors

Overall, a very good review on NGF and autoimmunity. The authors clearly have great knowledge about both fields of neuroscience and immunology. 

Please see the specific comments below:

1. In general, since this is a review, I don't think there is suppose to be a methods/results section in both the abstract and manuscript. Is this the format the journal wants reviews to be written? 

2. Line 19: For the abstract, please put down nerve growth factor (NGF) and state what any other abbreviation is within the abstract and manuscript. 

3. Line 50: Can you provide any references on how NGF regulates T and B cell differentiation? I never have heard of this and would love to see some papers about it.

4. Lines 58-78: If not requested by the journal, this materials and methods section is useless and needs to be cut out.

5. Line 97: Just curious, is this related to LRRK2 in anyway? 

6. For Figure 1, could the authors be more clear on what each of the signaling factors that NGF controls contributes towards the phenotypes they are mentioning from the innate immune response and the adaptive immune response? For example, out of all the listed immune modulations, what would NF-kappaB do to T cells, B cells, Mast cells, etc. Instead of just a big arrow pointing to all of the cells and suggesting Ras, PI3-kinase, phospholipase C-gamma1, NF-kappaB, and Jun kinase all do the same thing, can the authors separate out each of these signaling pathways and point towards which immune cells it affects and how it affects them? This way, it would be more clear on the precise pathway that NGF affects signaling pathways and those signaling pathways affect specific immune cell functionality. Also, can the authors explain in more details how TrkA,B,C competes with p75NTR for binding of NGF? Are both receptors found together on the same cell, or do some cells have Trk and some have p75NTR? Perhaps another figure going into more detail about the molecular components of the receptors TrkA,B,C and p75NTR would be a good figure to have. 

7. Lines 139-140: What do the authors think about why NGF is being produced by CD4+ T cells? Can the authors explain briefly perhaps on why T cells would need to produce NGF? Is it also found within the periphery as well, or are these T cells that produce NGF only found in the brain? 

8. Lines 141-147: Do B cells also produce NGF?

9. Lines 162-163: What do you mean by "our organism?" 

10. Lines 163-165: I don't think this means anything about cytokine functionality. This to me means that potentially NGF has an epitope very similar to that of HSV. See the term "molecular mimicry." Perhaps you can explain a little more about reference 52 and how they came up with the conclusion about cytokine functionality and NGF? 

11. Lines 173-175: Can the authors explain more in detail what the autoantibodies could also bind on, i.e., which cytoskeletal proteins and which portions of DNA can it bind onto?

12. Table 1 seems kind of compressed some, but this might be the format of the manuscript version the editor sent to the reviewers. But, if possible, could the authors either increase the width of the table or perhaps reduce the text size? 

13. What do the authors think about that autoantibodies against NGF cause problems, but anti-NGF could relief autoimmune diseases like arthritis? Wouldn't both autoantibody and anti-NGF cause the same modulations? Are the epitopes that both the antibodies bind onto different? Maybe some discussion of this should be added. 

14. Do the authors know if there are any genetic mutations that lead towards any modification of NGF production and/or Trk and/or p75NTR? If so, have these mutations been associated with autoimmune diseases? It would be really great to talk about  in the discussion the connection with genetics causing modulations with NGF, Trk, and p75NTR and if they are associated with the previously mentioned diseases. As autoimmune diseases have both a genetic and environmental component, it would be great to try to connect that with NGF, Trk and/or p75NTR mutations or environmental triggers that modulate these components. 

Author Response

Answers to the comments raised by Reviewer 2

Overall, a very good review on NGF and autoimmunity. The authors clearly have great knowledge about both fields of neuroscience and immunology.

Reply: we thank the reviewer for the positive comments. Indeed, according to the comments of the 2 reviewers we made changes highlighted in light yellow. Furthermore, we included an additional Figure and an additional Table.

Please see the specific comments below:

  1. In general, since this is a review, I don't think there is suppose to be a methods/results section in both the abstract and manuscript. Is this the format the journal wants reviews to be written?

Reply: As requested, we removed the methods section.

  1. Line 19: For the abstract, please put down nerve growth factor (NGF) and state what any other abbreviation is within the abstract and manuscript.

Reply: According to this comment, we made appropriate modifications in the abstract (please see the modified text highlighted in light yellow).

  1. Line 50: Can you provide any references on how NGF regulates T and B cell differentiation? I never have heard of this and would love to see some papers about it.

Reply: As suggested, we added a further Table (now Table 1 of the revised paper) to better describe this issue.

  1. Lines 58-78: If not requested by the journal, this materials and methods section is useless and needs to be cut out.

Reply: as suggested, we deleted this section

  1. Line 97: Just curious, is this related to LRRK2 in anyway?

Reply: according to this comment, we provided evidence showing a potential role of neurotrophins such as BDNF in patients with mutations of LRRK2 in neurotrophin-related diseases (page 2, lines 94-96 of the revised paper).

  1. For Figure 1, could the authors be more clear on what each of the signaling factors that NGF controls contributes towards the phenotypes they are mentioning from the innate immune response and the adaptive immune response? For example, out of all the listed immune modulations, what would NF-kappaB do to T cells, B cells, Mast cells, etc. Instead of just a big arrow pointing to all of the cells and suggesting Ras, PI3-kinase, phospholipase C-gamma1, NF-kappaB, and Jun kinase all do the same thing, can the authors separate out each of these signaling pathways and point towards which immune cells it affects and how it affects them? This way, it would be more clear on the precise pathway that NGF affects signaling pathways and those signaling pathways affect specific immune cell functionality.

Reply: As suggested, a further Table (Table 1 of the revised paper) on the role of these pathways in the immune cells was included in the revised paper.

Also, can the authors explain in more details how TrkA,B,C competes with p75NTR for binding of NGF? Are both receptors found together on the same cell, or do some cells have Trk and some have p75NTR? Perhaps another figure going into more detail about the molecular components of the receptors TrkA,B,C and p75NTR would be a good figure to have.

Reply: As requested, we added further knowledge on the competitive and cooperative mechanisms of TRK and p75NTR for binding NGF (pages 2 and 3, lines 97-117 of the revised paper).

  1. Lines 139-140: What do the authors think about why NGF is being produced by CD4+ T cells? Can the authors explain briefly perhaps on why T cells would need to produce NGF? Is it also found within the periphery as well, or are these T cells that produce NGF only found in the brain?

Reply: As suggested, other information was reported in the new Figure 2 on page 10 of the revised manuscript.

  1. Lines 141-147: Do B cells also produce NGF?

Reply: According to the request of the reviewer, we reported some evidence about this with the references (page 5, lines 173-186 of the revised paper).

  1. Lines 162-163: What do you mean by "our organism?"

Reply: we do apologize for the misunderstanding. This sentence was revised (page5, lines 187-188 of the revised paper).

  1. Lines 163-165: I don't think this means anything about cytokine functionality. This to me means that potentially NGF has an epitope very similar to that of HSV. See the term "molecular mimicry." Perhaps you can explain a little more about reference 52 and how they came up with the conclusion about cytokine functionality and NGF?

Reply: Indeed, this could be the case. We have further explained the study in question and have mentioned the possibility of molecular mimicry (pages 5 and 6, lines 190-206 of the revised text).

  1. Lines 173-175: Can the authors explain more in detail what the autoantibodies could also bind on, i.e., which cytoskeletal proteins and which portions of DNA can it bind onto?

Reply: this is a good question that should be investigated deeply.

  1. Table 1 seems kind of compressed some, but this might be the format of the manuscript version the editor sent to the reviewers. But, if possible, could the authors either increase the width of the table or perhaps reduce the text size?

Reply: The table was reformatted by the editors. Anyway, we reformatted the 2 Tables of the revised manuscript.

  1. What do the authors think about how autoantibodies against NGF cause problems, but anti-NGF could relieve autoimmune diseases like arthritis? Wouldn't both autoantibody and anti-NGF cause the same modulations? Are the epitopes that both the antibodies bind to different? Maybe some discussion of this should be added.

Reply: As suggested, more info was provided in the revised discussion of the paper (page 14, lines 530-545, of the revised text).

  1. Do the authors know if there are any genetic mutations that lead towards any modification of NGF production and/or Trk and/or p75NTR? If so, have these mutations been associated with autoimmune diseases? It would be really great to talk about in the discussion the connection with genetics causing modulations with NGF, Trk, and p75NTR and if they are associated with the previously mentioned diseases. As autoimmune diseases have both a genetic and environmental component, it would be great to try to connect that with NGF, Trk and/or p75NTR mutations or environmental triggers that modulate these components.

Reply: According to this crucial point, we added this hypothesis in the discussion (page 14, lines 562-566 of the revised text).

Round 2

Reviewer 2 Report

Comments and Suggestions for Authors

Overall, the authors answered all of my questions and added in my suggestions. 

The only problem now is some of the English and formatting issues.

1. For the abstract, remove the Methods section and the Background, Results, and Conclusion subheadings. As a review, all of this is essentially background. You can keep the Conclusion section if you want.

2. The paper is sectioned into: 1. Introduction, 3. Neurotrophins, 4. Neurotrophins and the immune system....etc. Since the "Methods" section was removed, the rest of the paper was not re-numbered properly. 

3. Make sure all gene names are italicized, such as LRRK2

4. Lines 97-117 could use more references. Or, is this section of the text only based on References 50 and 51?

5. For Table 1, could the references for each claim be put into the table? 

6. Lines 190-206 need references, unless again, is this all based on Reference 64? 

7. The figure text in Figure 2 is way to long. Could the authors summarize this better or put some of the figure text into the main text? Great figure still! 

8. Again, references need to be added...look at lines 530-545. 

9. Lastly, have someone proficient in English go over the entire paper very carefully one more time.

Other than that, good job! Great paper! 

Comments on the Quality of English Language

Overall grammar, sentence structure, and spelling needs to be looked over one more time. 

Author Response

Overall, the authors answered all of my questions and added in my suggestions. 

We sincerely thank the reviewer for the helpful comments aimed at improving the quality of the paper.

The only problem now is some of the English and formatting issues.

  1. For the abstract, remove the Methods section and the Background, Results, and Conclusion subheadings. As a review, all of this is essentially background. You can keep the Conclusion section if you want.

Reply: As suggested, the abstract was revised.

  1. The paper is sectioned into: 1. Introduction, 3. Neurotrophins, 4. Neurotrophins and the immune system... etc. Since the "Methods" section was removed, the rest of the paper was not re-numbered properly. 

Reply: we apologize for the awful typing mistakes. Now the sections are correctly numbered sequentially.

  1. Make sure all gene names are italicized, such as LRRK2

Reply: As suggested, we made appropriate changes (see page 2 line 92; page 8 line 243, page 13 line 496)

  1. Lines 97-117 could use more references. Or, is this section of the text only based on References 50 and 51?

Reply: According to this comment, further refs were included in the paper.

  1. For Table 1, could the references for each claim be put into the table? 

Reply: As suggested, we added additional references to the tables in a third column.

  1. Lines 190-206 need references, unless again, is this all based on Reference 64? 

Reply: As requested, appropriate refs were added (now lines 184-200).

  1. The figure text in Figure 2 is way to long. Could the authors summarize this better or put some of the figure text into the main text? Great figure still! 

Reply: As suggested, we summarized the figure caption as requested.

  1. Again, references need to be added...look at lines 530-545. 

Reply: we added references as requested by the reviewer (now lines 513-526).

  1. Lastly, have someone proficient in English go over the entire paper very carefully one more time.

Reply: we revised the manuscript and especially the new sentences in order to ameliorate the language.

Other than that, good job! Great paper! 

Reply: again, many thanks

Comments on the Quality of English Language

Overall grammar, sentence structure, and spelling needs to be looked over one more time. 

Reply: we made efforts to improve the English quality of the paper.